

# Revealing Seasonal Plasticity of Whole-Plant Hydraulic Properties Using Sap-Flow and Stem Water-Potential Monitoring

Zhechen Zhang[1], Huade Guan[1*], Erik Veneklaas[2], Kamini Singha[3], and Okke Batelaan[1]

[1]National Centre for Groundwater Research and Training, College of Science & Engineering, Flinders University, Adelaide, South Australia, Australia.
[2]School of Biological Sciences, The University of Western Australia, Perth, Western Australia, Australia
[3]Department of Geology & Geological Engineering, Hydrologic Science and Engineering Program, Colorado School of Mines, Golden, CO, USA.

*Corresponding author: Huade Guan (huade.guan@flinders.edu.au)*

**Abstract.** Plant hydraulic properties are critical to predicting vegetation water use as part of land-atmosphere interactions and plant responses to drought. However, current measurements of plant hydraulic properties are labour-intensive, destructive, and difficult to scale up, consequently limiting the comprehensive characterization of whole-plant hydraulic properties and hydraulic parameterization in land-surface modelling. To address these challenges, we develop a method, a pumping-test analogue, using sap-flow and stem water-potential data to derive whole-plant hydraulic properties, namely maximum hydraulic conductance, effective capacitance, and $\Psi_{50}$ (water potential at which 50% loss of hydraulic conductivity occurs). Experimental trials on *Allocasuarina verticillata* indicate that the parameters derived over short periods (around 7 days) exhibit good representativity for predicting plant water use over at least one month. We applied this method to estimate near-continuous whole-plant hydraulic properties over one year, demonstrating its potential to supplement existing labour-intensive measurement approaches. The results reveal the seasonal plasticity of the effective plant hydraulic capacitance. They also confirm the seasonal plasticity of maximum hydraulic conductance and the hydraulic vulnerability curve, known in the plant physiology community while neglected in the hydrology and land-surface modelling community. It is found that the seasonal plasticity of hydraulic conductance is associated with climate variables, providing a way forward to represent seasonal plasticity in models. The relationship between derived maximum hydraulic conductance and $\Psi_{50}$ also suggests a trade-off between hydraulic efficiency and safety of the plant. Overall, the pumping-test analogue offers potential for better representation of plant hydraulics in hydrological modelling, benefitting land-management and land-surface process forecasting.

## 1 Introduction

Plant hydraulic properties, such as maximum xylem hydraulic conductance, vulnerability to cavitation, and maximum stomatal conductance, are fundamental plant functional traits regulating hydraulic processes in plants. They play critical roles in regulating transpiration and growth by controlling plant water uptake (Anderegg and Meinzer, 2015; Matheny et al., 2017a), and are instrumental in predicting plant water storage (Huang et al., 2017), and drought-driven tree mortality risk (Liu et al., 2017; Powell et al., 2017; Torres-Ruiz et al., 2024).

Hydraulic properties of different species and plant communities reflect different water-use strategies, which determine various responses to root-zone moisture conditions (Matheny et al., 2017b; Barros et al., 2019). Consequently, plant hydraulic properties are critical to predicting plant water use as part of land-atmosphere interactions and ecosystem responses to drought.

Plant hydraulic properties exhibit temporal variability. Variations can be expected across ontogeny (Mencuccini, 2002), between seasons (Jacobsen et al., 2007), and in response to hydroclimatic events such as droughts (Anderegg and Callaway, 2012). For example, the maximum hydraulic conductance can change due to carbon allocation to stem growth (Buckley and Roberts, 2006; Potkay et al., 2021). Some plant physiology studies have demonstrated that plant hydraulic properties have seasonal plasticity, which means they are time-variant with



time of year. Past studies have reported that the maximum hydraulic conductance and the $\Psi_{50}$ (xylem water

potential at which 50% loss of hydraulic conductivity occurs) values of several tree species vary significantly in different seasons (e.g., *Artemisia tridentata* (Kolb and Sperry, 1999), several Californian species (Jacobsen et al., 2007), several Mediterranean species (Sorek et al., 2022), and *Pinus halepensis* (Feng et al., 2023)). Similarly, Li et al. (2023) reported that most of the leaf hydraulic traits of Korean pine and spruce significantly changed over four months. Ecological modelers found that the hydraulic properties calibrated in one season are not

transferable to other seasons (Steppe et al., 2008; Baert et al., 2014; Salomón et al., 2017), which likely due to the plasticity of plant hydraulic properties. Despite the evidence of seasonal plasticity of hydraulic properties, it is not known yet how common this phenomenon is and what mechanisms drive it (Feng et al., 2023), in part become of limited high-frequency, seasonal measurements. Although continuous monitoring of plant water use has become more common, very few studies have combined sap flux and water status measurements, which

may be key to informing on the seasonal variation in plant hydraulics.

Despite evidence of seasonal plasticity in plant hydraulic properties, they are largely treated as constant parameters in both single-plant models (Bohrer et al., 2005; Christoffersen et al., 2016; Deng et al., 2017) and ecosystem-scale models (Kennedy et al., 2019; Li et al., 2021), due to the complexity of parameterizing time-variant properties. There have been a few attempts to consider plant hydraulics in land-surface models (Kennedy

et al., 2019; Li et al., 2021; Xie et al., 2023; Paschalis et al., 2024); although these efforts have significantly improved model performance, none of the models utilize time-variant hydraulic properties. To facilitate time-invariant parameterization, some models have simplified the structure of their hydraulic module; for example, the latest Noah-MP-PHS model includes only hydraulic conductance and excludes hydraulic capacitance (Li et al., 2021). The work of Jiménez-Rodríguez et al. (2024) suggests that parameterization of maximum hydraulic

conductance is an important yet unresolved issue in land-surface models such as CLM. While time-invariant parameterization of plant hydraulics in models is challenging, time-variant parameterization is even more difficult.

Current methods for estimating plant hydraulic properties are one limitation on parameterization of plant hydraulics in models. The most commonly used measurements of hydraulic properties involve collecting

'snapshots' in time (for example, dehydration methods to measure hydraulic conductivity) (Sperry et al., 1988; Zhang et al., 2018). These 'snapshot' methods are typically conducted in the laboratory using stem sections collected from the field. They are destructive and labor-intensive, resulting in static and limited data (Novick et al., 2022) that cannot capture seasonal variation. Furthermore, hydraulic properties measured at stem scales are difficult to scale up to representative whole-plant parameter values required for modeling. Few studies have

investigated whole-plant hydraulic properties based on field measurements (Zeppel et al., 2008; Deng et al., 2017), and such early attempts still adopted time-invariant hydraulic vulnerability curves. Due to data limitations, applying hydraulic properties of specific species in a land-surface model is quite challenging. Most studies use plant hydraulic properties based on the Plant Functional Type (PFT) classification network (Paschalis et al., 2024; Raghav et al., 2024). Including different species within the same PFT classification

neglects inter- and intra-species variation in hydraulic properties. The static and limited plant hydraulic properties from current lab-based methods hinder the ability to capture temporal dynamics, whole-plant representation, and intra- and inter-species variability.



Model-data fusion methods have recently shown potential to address challenges such as inter- and intra-species variability and whole-plant representativeness (Liu et al., 2020b; Lu et al., 2022), but they still cannot capture the temporal variability of hydraulic properties. Such approaches, combined with a given model, estimate the properties that best match the observed temporal variation of evapotranspiration or sap flow. Liu et al. (2021) derived ecosystem-scale plant traits (stomatal conductance and $\Psi_{50}$) across the globe using a model-data fusion approach constrained by remote-sensing products of evapotranspiration, vegetation optical depth, and soil moisture. Building upon this foundation, Lu et al. (2022) estimated species-specific hydraulic properties based on sap flow measurements. Although these model-data fusion methods have shown promise in enhancing our understanding of spatial and intra- and inter-species variations of hydraulic properties, at the whole-plant scale, quantifying temporal variation of plant hydraulics remains a challenge because of the extensive data inputs needed for model-data fusion methods.

This paper aims to develop a new method to estimate plant hydraulic properties in the field continuously and non-destructively. Specifically, we quantify whole-plant hydraulic conductance and capacitance using sap-flow and stem water-potential data. We test the new proposed method on several drooping sheoak trees and investigate the seasonal variation in hydraulic properties of this species.

## 2 Methodology

### 2.1 Pumping-test analogue theory development

We develop a pumping-test analogue to estimate time-variant plant hydraulic properties, borrowing from the concept of pumping tests in hydrogeology where the hydraulic properties of an aquifer is estimated based on hydraulic head responses (measured at observation wells) to the disburbance induced at the pumping well. In the soil-plant continuum, root water uptake and transpiration introduce the hydraulic disturbance, resulting in simultaneous plant water-potential responses. Therefore, by measuring the "pumping" — dynamic transpiration (or sap flow) and the "corresponding response", i.e., the plant water-potential change, and relating these two aspects through a physically based model, we can derive the key hydraulic properties that govern plant hydraulic processes, such as plant hydraulic conductance and capacitance. In addition, plants also work as hydraulic 'capacitors' that temporally store and release water depending on the water demand from the canopy and water uptake from the root zone, resulting in diel fluctuation of plant water potential. With an increasing number of long-term observations of sap flow and an increasing ability to measure plant water potential in recent years (Restrepo-Acevedo et al., 2024), the approach proposed here is increasingly feasible, providing the necessary data to link water-flux changes with plant water potential and derive key hydraulic properties.

Here, we use a whole-plant hydraulics model to connect measured water flux and water-potential changes. An analogue resistance-capacitance model (RC model) is commonly used to simulate hydraulic processes in the soil-plant-atmosphere continuum, with two parts: plant water uptake flux and storage change flux. There are a number of RC models of different complexity (Loustau et al., 1998; Cowan, 1965; Steppe et al., 2006; Salomón et al., 2017). The whole-plant RC model (Liu et al., 2021) has the simplest structure with only one circuit (conceptualized model shown in Fig. 1), as follows:



$$E_C = Q - \frac{dS_P}{dt} \tag{1}$$

where $E_C$ (cm$^3$ cm$^{-2}$ h$^{-1}$) is transpiration flux density, Q (cm$^3$ cm$^{-2}$ h$^{-1}$) is plant water uptake flux density and $\frac{dS_P}{dt}$ (cm$^3$ cm$^{-2}$ h$^{-1}$) is the whole-plant-equivalent transpirable-storage change rate. All flux densities are normalized over the sapwood area. Plant water uptake flux density Q is calculated by a Darcy's law equivalent formulation between two effective nodes, the plant and root-zone nodes (Steppe et al., 2006):

$$Q = k_P * (\psi_{rz} - \psi_P) \tag{2}$$

where $k_P$ (cm$^3$ cm$^{-2}$ h$^{-1}$ MPa$^{-1}$) is the whole-plant hydraulic conductance, $\psi_P$ (MPa) is the effective plant water potential to represent the water status of the plant node, and $\psi_{rz}$ (MPa) is the bulk root-zone water potential.

The hydraulic capacitance $C_P$ (cm$^3$ cm$^{-2}$ MPa$^{-1}$) is defined as the ratio of the change in the amount of water storage to the change in water potential (Steppe et al., 2006; Salomón et al., 2017; Hunt et al., 1991):

$$C_P = \frac{dS_P}{d\psi_{P_S}}. \tag{3}$$

where $\psi_{P_S}$ (MPa) is the effective water potential at which stored water is held by the plant. Substituting Eq. 2 and Eq. 3 into Eq. 1, the transpiration flux density Ec can be calculated as:

$$E_C = k_P * (\psi_{rz} - \psi_P) - C_P * \frac{d\psi_{P_S}}{dt}. \tag{4}$$

The whole-plant conductance varies with effective whole-plant water potential, following a Weibull shape
vulnerability curve (Sperry et al., 1998; Deng et al., 2017):

$$k_P = k_{max} e^{-\left(-\frac{\psi_P}{d}\right)^c} \tag{5}$$

where $k_{max}$ (cm$^3$ cm$^{-2}$ h$^{-1}$ MPa$^{-1}$) is the maximum hydraulic conductance, and $d$ and $c$ are two curve-fitting parameters.

Plant hydraulic conductivity is not only affected by water potential (Eq.5) but also by temperature (Yang et al.,
2020; Cochard et al., 2000). Hydraulic conductivity is proportional to the permeability of the porous material and inversely proportional to the viscosity (Marshall et al., 1996). We assume that the permeability in a root-zone and plant continuum is constant over a short time period, as it is only related to the medium structure. Viscosity, meanwhile, is strongly dependent on the temperature and type of fluid (Marshall et al., 1996), so hydraulic conductance at one specific temperature can be expressed as follows:

$$k_{P,T} = k_{P,T_{ref}} * \frac{\eta_{T_{ref}}}{\eta_T} \tag{6}$$

where $k_{P,T}$ and $k_{P,T_{ref}}$ are hydraulic conductance at specific temperature $T$ and at reference temperature $T_{ref}$, respectively, and $\eta_T$ and $\eta_{T_{ref}}$ are water viscosity at specific temperature $T$ and at reference temperature $T_{ref}$, respectively. $T_{ref}$ is 25 °C in this study. The empirical equation used in this study of water viscosity variation with temperature is as follows (Heggen, 1983; Dingman, 2015):

$$\eta_T = 2.0319 * 10^{-4} + 1.5883 * 10^{-3} * exp\left[-\left(\frac{T^{0.9}}{22}\right)\right] \tag{7}$$

where $T$ is in °C.

The whole-plant hydraulic capacitance also varies with water potential. We adopt the following relation:

$$C_P = C_{max} \left(\frac{\psi_0 - \psi_{P_S}}{\psi_0}\right)^{-p} \tag{8}$$



where $C_{max}$ (cm$^3$ cm$^{-2}$ MPa$^{-1}$) is the maximum hydraulic capacitance, and $\psi_0$ (MPa) and $p$ are empirical

parameters (Chuang et al., 2006).

Together, Equations 4-8 define the complete whole-plant hydraulic model. There are three state variables ($\psi_P$, $\psi_{P_S}$, and $\psi_{rz}$) and six parameters ($k_{max,25}$, $d$, $c$, $C_{max}$, $\psi_0$, and $p$), all of which are being estimated by this method. Although the conceptualized effective plant water potential $\psi_P$ cannot be measured, water potential in a part of the plant, e.g., the stem, can be monitored using stem psychrometers, which covary with whole-plant

effective water potential. Therefore, we assume that the effective plant water potential can be approximated by the stem water potential measured by psychrometer at a specific location. Similarly, the assumption that the water potential of plant water storage $\psi_{P_S}$ is equivalent to the observed stem water potential is a reasonable approximation (Liu et al., 2021).

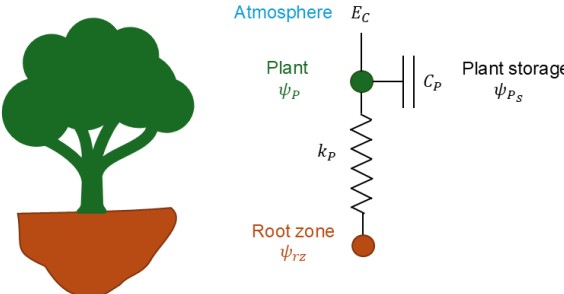

**Figure 1: Conceptual diagram of the whole-plant hydraulics model.**

## 2.2 Site and data description

Data collected from three drooping sheoak trees (*Allocasuarina verticillata*) were used in this study to validate the proposed pumping-test analogue method. They grow on a hillslope at the Bedford Park Campus of Flinders University (35°1′49″ S, 138°34′28″ E). This site experiences a Mediterranean climate characterized by distinct wet and dry seasons throughout the year. Stem water potential was measured by thermocouple psychrometers

(PSY1, ICT International Pty Ltd., Australia) at half-hourly intervals. Sap-flow rates were measured at half-hourly intervals at ~1.2 m height of the trunk by heat-pulse sap flow meters (SFM1, ICT International Pty Ltd., Australia). Figure 2 shows the hourly stem water potential and sap-flow rate recordings. Geometric data of the trees and detailed measurement settings were reported in Luo et al. (2020). The tree numbers are the same as in Luo et al. (2020). However, only Trees 2, 3, and 4 are used in this study, as the sap-flow and stem water-

potential recordings of Tree 1 were too short for this study. Air temperature and radiation data (in Fig.3a and 3b) were from a weather station located at the Bedford Park Campus. Precipitation data (in Fig.3c) were from a nearby weather station of Kent Town (34.92° S, 138.62°E).





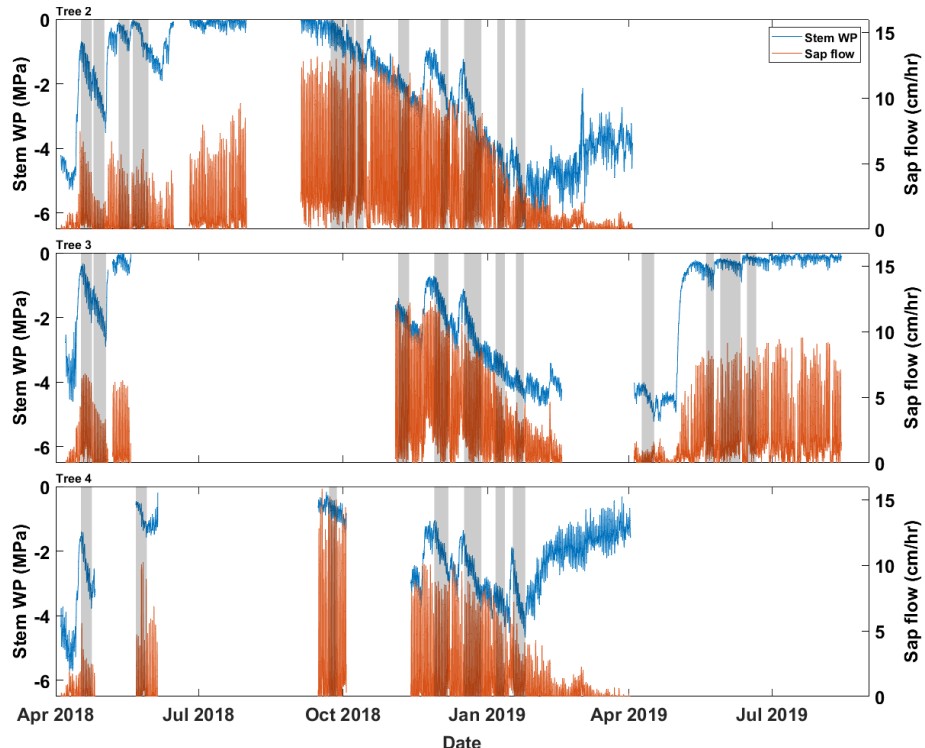

**Figure 2: The input data for the pumping-test analogue on three drooping sheoak trees. Blue lines show the hourly stem water-potential recordings and orange lines are the hourly sap-flow rate. The grey periods are the selected calibration periods for each tree, which are periods without any rainfall.**

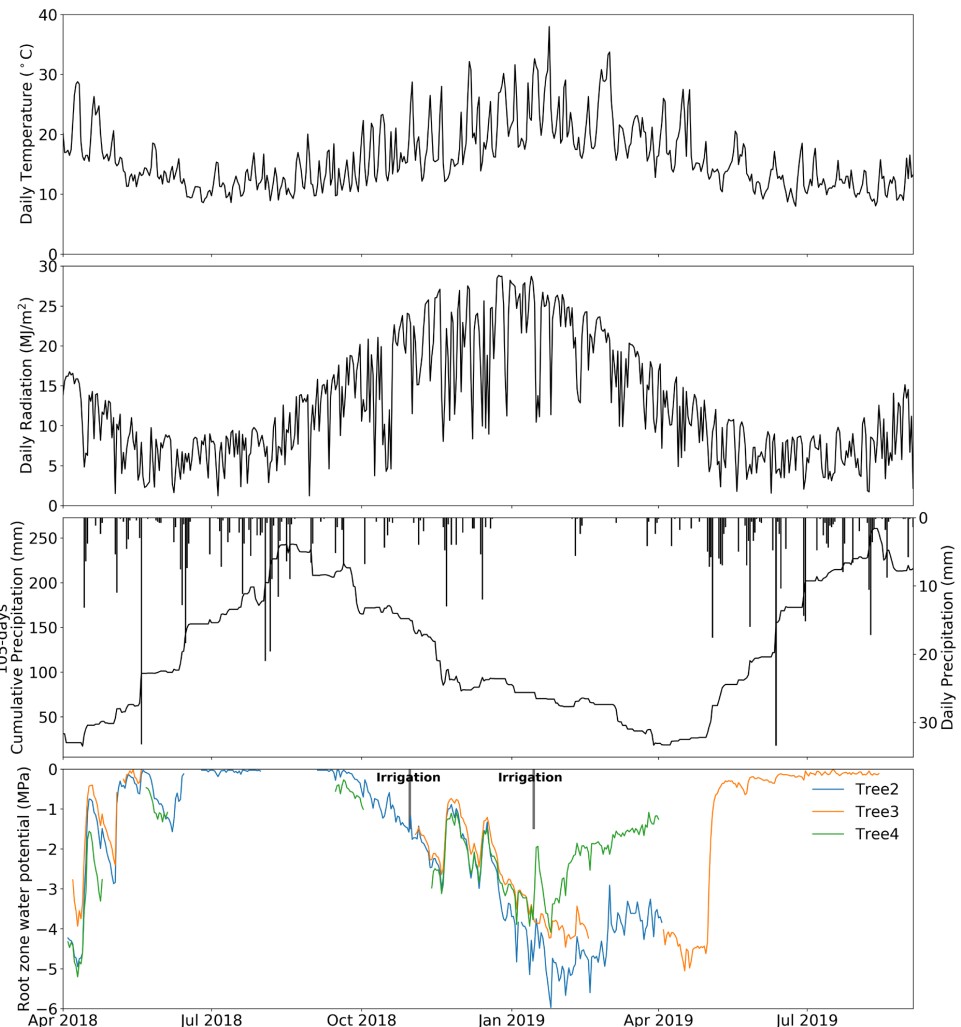

**Figure 3: (a) Daily air temperature, (b) radiation, (c) daily precipitation and the antecedent 105-day cumulative precipitation, and (d) the estimated root zone water potential derived from stem water potential of each tree. The irrigation experiments are indicated in panel (d). Two drip irrigation events (not shown in panel (c)) were conducted during the study period. The first occurred on 30 October 2018, applying 80 L of water to Tree 2. The second took place between the late afternoon of 14 January and 15 January 2019, applying 100 L of water to Tree 2 and Tree 4, respectively.**

## 2.3 MCMC Implementation

A Markov Chain Monte Carlo (MCMC) method was used to infer the parameters of the whole-plant hydraulics model because it has been successfully applied for deriving whole-plant hydraulic traits (Liu et al., 2021; Lu et al., 2022), as well as for estimating plant hydraulic model parameters (Deng et al., 2017). We used an MCMC sampling scheme within the DiffeRential Evolution Adaptive Metropolis (DREAM) algorithm (Vrugt et al., 2009). The MCMC chain number was set to 10 and the iteration steps were set to 3000 times per parameter (18000 in total). We took the last 25% of samples as posterior samples, which is a common setting in DREAM





applications on hydrological problems (Baig et al., 2022). The objective function we set is the sum of squared errors (SSE) between the observed and simulated sap flux density (cm/h). The parameters of the whole-plant hydraulic model and their calibration range are shown in Table 1. Since the pumping-test analogue is based on plant hydraulic processes, the hydraulic properties it estimates represent the effective hydraulic properties contributing to transpiration rather than the hydraulic properties measured in the lab. $k_{max,25}$ and $C_{max}$ correspond to maximum effective hydraulic conductance and capacitance, respectively. All ranges were set as large as reasonable to minimize the chance of missing the global minimum in the optimization. The range of empirical parameters of the capacitance curve was based on the literature (Chuang et al., 2006; Deng et al., 2017; Silva et al., 2022). To minimize the effect of measurement error of stem water potential, we applied a one-hour moving average to the raw measured stem water potential. The predawn bulk root-zone water potential $\psi_{rz,predawn}$ (Fig.3d) was approximated by the measured stem water potential, based on the predawn equilibrium assumption (Sperry et al., 2002; Wang et al., 2014; Liu et al., 2020a). Predawn was defined as 3:00 am to 5:00 am in this study, and bulk root-zone water potential $\psi_{rz}$ at other times (Fig.3d) was estimated by interpolation between each pair of predawn estimation $\psi_{rz,predawn}$.

**Table 1: Parameters of the whole-plant hydraulic model and calibration range.**

| Symbol | Description | Range |
|---|---|---|
| $k_{max,25}$ (cm/h/MPa) | Maximum effective hydraulic conductance at 25 °C | 50-500 |
| $c$ (-) | Conductivity curve-fitting parameter | 1.5-5[*] |
| $d$ (-) | Conductivity curve-fitting parameter | 1.5-5[*] |
| $C_{max}$ (cm/MPa) | Maximum effective hydraulic capacitance | 0-500 |
| $\psi_0$ (MPa) | Empirical parameters of capacitance curve | 0-5 |
| $p$ (-) | Empirical parameters of capacitance curve | 1.5-5 |

*These fitting parameters are shown in Equation 5. The low boundaries of the conductance-curve fitting parameters were set up as 1.5 to ensure that the conductance curve was S-shaped.*

**2.4 Pumping-test analogue validation**

First, we validated that the proposed pumping-test analogue framework could derive reasonable hydraulic properties based on sap-flow and stem water-potential data. Given that we interpolated sub-daily bulk root-zone water-potential values between each predawn measurement, precise estimation of bulk root-zone water potentials was hindered in cases of daytime wetting events. To mitigate this issue, we excluded such days and selected periods with at least five days of consecutive predawn stem water-potential reductions as the calibration periods (grey boxes in Fig.2). For example, there were twelve calibration periods for Tree 2 (P1 – P12). The validation periods were not predetermined but rather dynamically extended from each calibration period. This extension proceeded at a rate of one hour per step in both temporal directions until it encompassed the entirety of the dataset. This approach offered a flexible way to assess the representativeness of the properties derived during the calibration period across different times of the year.

We used three criteria to filter the derived hydraulic properties for further seasonal variation analysis. The first criterion was that the Nash-Sutcliffe Efficiency (NSE) must be higher than 0.6 in the calibration period. The second was that the extended validation period with NSE higher than 0.6 must be longer than 20 days. In addition to the calibration and validation criteria, ensuring the convergence of posterior samples was another vital factor to control the reliability and stability of parameter estimation. The third criterion was that the



Gelman and Rubin statistics (R_stat) must be lower than 1.2 to confirm the convergence of the MCMC process (Gelman and Rubin, 1992; Deng et al., 2017).

### 2.5 Deriving near-continuous plant hydraulic properties

After successfully validating that the pumping-test analogue framework could derive representative plant hydraulic properties using short-term data, we applied this method to obtain near-continuous plant hydraulic
properties for analysing the relationships between the properties ($k_{max,25}$, $\Psi_{50}$, and $C_{max}$) and climate variables. In this study, we selected three key climate variables that are commonly known to influence plants: radiation, temperature, and precipitation. Unlike the validation phase, the actual application did not require selecting no-rain periods as calibration periods. Instead, we used a dynamic-window approach to estimate plant hydraulic properties every day. The influence of rainfall on root-zone water potential remained a concern; therefore, we
selected a 20-day duration for the dynamic window. Longer windows can help mitigate the influence of rainfall, because rain takes up less of the total data length. Additionally, the calibrated parameters had to meet two filter criteria: 1) the model must converge, and 2) NSE during the calibration period should be greater than 0.7. While we did not conduct validation, we tightened the criterion for the calibration period.

The plant hydraulic properties obtained using the dynamic-window method were analysed through multiple
linear regression to examine correlations with the corresponding climate variables. Radiation and temperature were represented by 20-day averages corresponding to the dynamic window, while rainfall was represented by the cumulative precipitation prior to the calibration period, with the cumulative days chosen based on optimal performance in the multiple linear regression model.

### 3 Results and Discussion

### 3.1 Pumping test analogue validation results

We successfully simulated the hourly sap-flow rate during the calibration periods. For most periods, the simulated sap flow in Tree 2 closely matched the observed patterns (Fig. 4, column 1), especially the diel variations. We even captured the sudden fluctuations caused by weather, such as radiation reduction due to cloud cover. For example, the calibrated whole-plant hydraulic model reproduced the sudden drops in sap flow
at daytime on May 15th (P3 in Fig. 4) and Oct 12th (P7). The NSEs for the Tree 2 calibration periods were higher than 0.8, except for P10 to P12. The simulated sap flow (orange line) had an obvious time lag compared with the observed data in the last periods.

The validation results show that the derived hydraulic properties have good representativeness in the weeks before and after the calibration periods. The hydraulic properties derived during autumn (P1 - P4) effectively
reproduced sap-flow rates during autumn and winter (April to July; Fig. 4, column 2). However, the results tended to underestimate sap-flow rates in spring and summer (September to February). Conversely, hydraulic properties derived from summer periods (P5 - P9) exhibited an opposing pattern, resulting in accurate estimation in spring and summer but overestimation during autumn and winter. This result indicates that the proposed pumping-test analogue method can derive hydraulic properties with robust seasonal representativeness and
capture the presence of seasonal variations in these properties as well. This result is consistent with previous studies in the plant physiology community, which found that the calibrated hydraulic properties in ecological



models are not transferable across seasons (Steppe et al., 2008; Baert et al., 2014; Salomón et al., 2017). The NSEs decreased with the extension of the validation period, as expected. The number of days when NSEs were higher than the threshold 0.6 was 50 days or more for most validation periods (Fig. 4, column 3). The validation
criterion was set to a required minimum extension of 20 days in the validation period, in addition to the calibration period of 5-7 days. This guaranteed that with a week of monitoring data, we can derive plant hydraulic properties that were representative for at least one month. The calibration and validation results of the other two trees confirmed the findings for Tree 2 (Fig. S1 & S2).

The derived properties for the three trees based on the three criteria are presented in Table 2. The derived
hydraulic properties were considered acceptable for over 50% of the calibration periods. The derived properties that did not meet the criteria were primarily concentrated in January and April; for example, in the period of January 6th, 2019, none of the three trees passed the filtering criteria. This is likely because the simple whole-plant hydraulic model fails under a severe water-stress condition. We have treated the plant as a single capacitor to simplify the model, but realistically, stems contain many capacitors distributed throughout wood and bark
with resistors between the wood and bark (Hölttä et al., 2006). Under extreme water stress, the hydraulic resistance between the xylem and phloem likely increases (Baert et al., 2014), effectively preventing the bark from contributing to total capacitance. Hence, capacitance dynamics can behave differently under severe stress than otherwise. The pumping-test analogue requires careful and critical validation under severe water stress. Moreover, it is worth noting that during periods of sufficient water conditions, such as winter in this study, the
stem psychrometers are prone to failing (Fig. 2). This may be due to water entering the psychrometer chamber from the ambient environment or sapwood in the wet season. The psychrometer measures water potential based on Peltier cooling; once there is water in the chamber, the stem psychrometer fails to work. New sensors, the microtensiometer (FloraPulse, Davis, CA, USA), may be more robust and improve the continuity of data (Black et al., 2020).







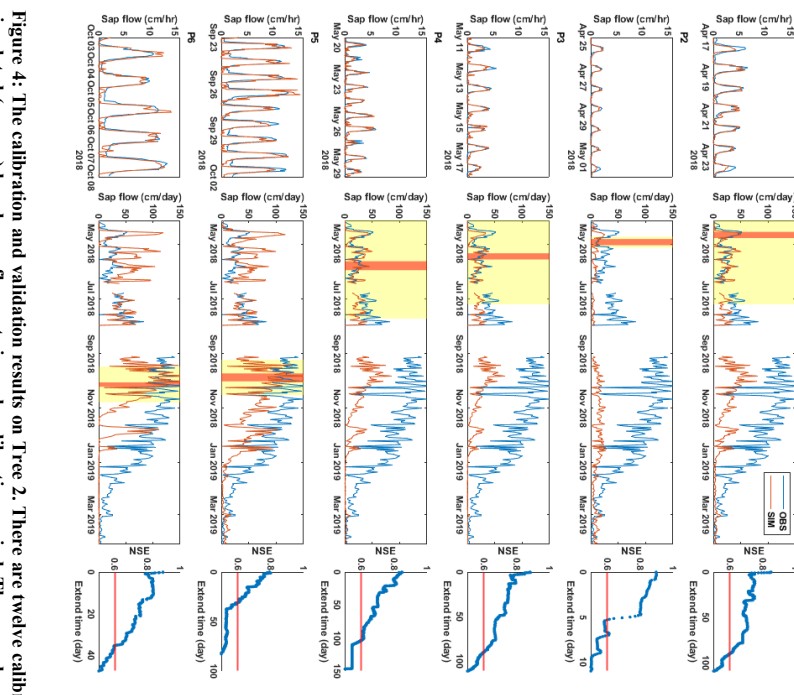

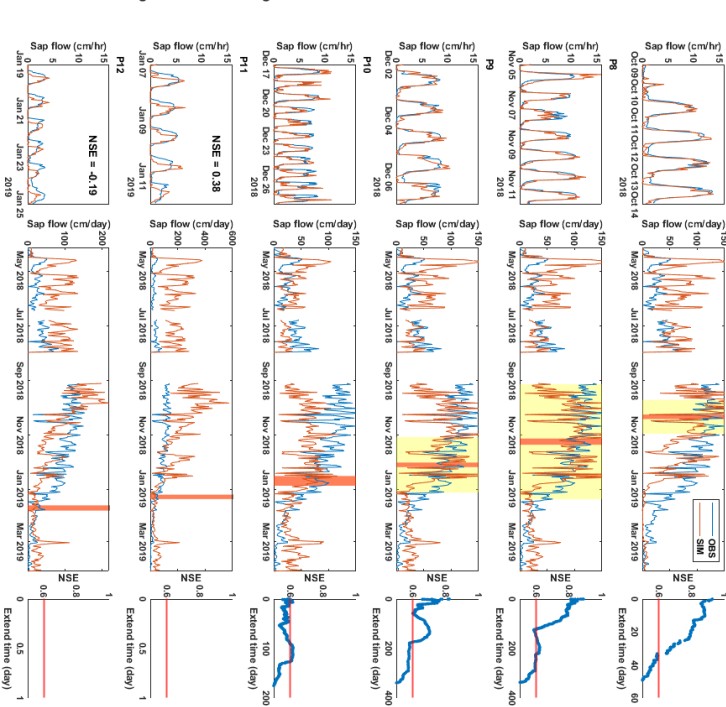

**Figure 4:** The calibration and validation results on Tree 2. There are twelve calibration periods (P1 – P12) in total. The first column for each period shows the observed (blue) and simulated (orange) hourly sap-flow rate in each calibration period. The second column for each period shows the observed (blue) and simulated (orange) daily sap-flow rate in the whole period. The orange boxes are the calibration periods, and the yellow boxes are the representative periods with validation NSE larger than 0.6. The third column for each period is the NSE changes with validation periods extending from the calibration periods. The orange line indicates the NSE threshold of 0.6.



**Table 2:** Filtered results showing whether the estimated hydraulic properties of each tree during specific periods met all three defined criteria. Green means that the hydraulic properties of the tree in that period passed the criterion; red means failing the criterion; grey means no measurement data in that period. Criteria 1 was that the NSE must be higher than 0.6 in the calibration period. Criteria 2 was that the extended validation period with NSE higher than 0.6 must be longer than 20 days. Criteria 3 was that the Gelman and Rubin statistics (R_stat) must be lower than 1.2 to confirm the convergence of the MCMC process.

| Tree | Criteria | P1 17/04/2018 | P2 25/04/2018 | P3 11/05/2018 | P4 20/05/2018 | P5 22/09/2018 | P6 3/10/2018 | P7 9/10/2018 | P8 5/11/2018 | P9 28/11/2018 | P10 17/12/2018 | P11 6/01/2019 | P12 19/01/2019 | P13 9/04/2019 | P14 20/05/2019 | P15 29/05/2019 | P16 15/06/2019 |
|---|---|---|---|---|---|---|---|---|---|---|---|---|---|---|---|---|---|
| Tree 2 | Criteria 1 | ✓ | ✓ | ✓ | ✓ | ✓ | ✓ | ✓ | ✓ | ✓ | ✓* | - | - | - | - | - | - |
|  | Criteria 2 | ✓ | ✗ | ✓ | ✓ | ✓ | ✓ | ✓ | ✓ | ✓ | ✓ | - | - | - | - | - | - |
|  | Criteria 3 | ✗ | - | - | - | ✓ | - | ✗ | ✓ | ✓ | ✓ | - | - | - | - | - | - |
| Tree 3 | Criteria 1 | ✓ | ✓ | ✓ | ✓ | ✓ | ✓ | ✓ | ✓ | ✓ | ✓ | ✗ | ✓ | ✗ | ✓ | ✓ | ✓ |
|  | Criteria 2 | ✓ | ✓ | ✓ | ✓ | ✓ | ✓ | ✓ | ✓ | ✗ | ✓ | ✓ | ✗ | - | ✓ | ✓ | ✗ |
|  | Criteria 3 | ✗ | ✓ | ✓ | ✓ | ✓ | ✓ | ✗ | ✓ | - | ✓ | ✗ | - | ✗ | ✓ | ✓ | - |
| Tree 4 | Criteria 1 | ✓ | - | - | ✓ | ✓ | - | - | - | ✓ | ✓ | ✓ | ✓ | - | - | - | - |
|  | Criteria 2 | ✗ | - | - | ✗ | ✓ | - | - | - | ✓ | ✓ | ✗ | ✗ | - | - | - | - |
|  | Criteria 3 | ✗ | - | - | - | ✓ | - | - | - | ✓ | ✓ | ✗ | - | - | - | - | - |

*   *P10 failed the second criterion, strictly, but because the validation NSEs were around 0.6 in more than 50 days, this period was included in the subsequent analysis.*



### 3.2 Seasonal variation of hydraulic conductance and capacitance

The pumping-test analogue reveales seasonal variations in plant hydraulic properties. The relationship between the estimated hydraulic conductance $k_P$ and plant water potential $\psi_P$ changed across calibration periods, and is therefore time-variant, rather than time-invariant (Fig. 5a). Figures 5b and 5c further illustrate the seasonal variations of two key parameters—maximum hydraulic conductivity ($k_{max,25}$) and $\Psi_{50}$—that control this relationship. The posterior distributions of the maximum hydraulic conductance are very narrow, indicating a low uncertainty of the derived properties (Fig. S3).

The three trees display the same pattern for the maximum hydraulic conductance $k_{max,25}$, with higher values in early spring (September), a gradual decrease from spring to summer (October to January), and a return to higher maximum hydraulic conductance in the following spring and early summer (June 2019) (Fig. 5b). This indicates that the maximum hydraulic conductance $k_{max,25}$ dropped as the drought stress intensified in the dry season and recovered with root zone moisture replenishment in wet season. Despite April and May being at the transition into the wet season, the maximum hydraulic conductance $k_{max,25}$ stayed relatively low. This result indicates that plants and their xylem embolisms need time to recover from the drought (or dry season). The three trees also display the same pattern for $\Psi_{50}$, which decreased from high values (low cavitation resistance) in spring to low values (high resistance) in the dry summer months of Dec-Jan (Fig.5c). This may reflect progressive loss of the most vulnerable conduits as soils dry out and tree water potentials decline.





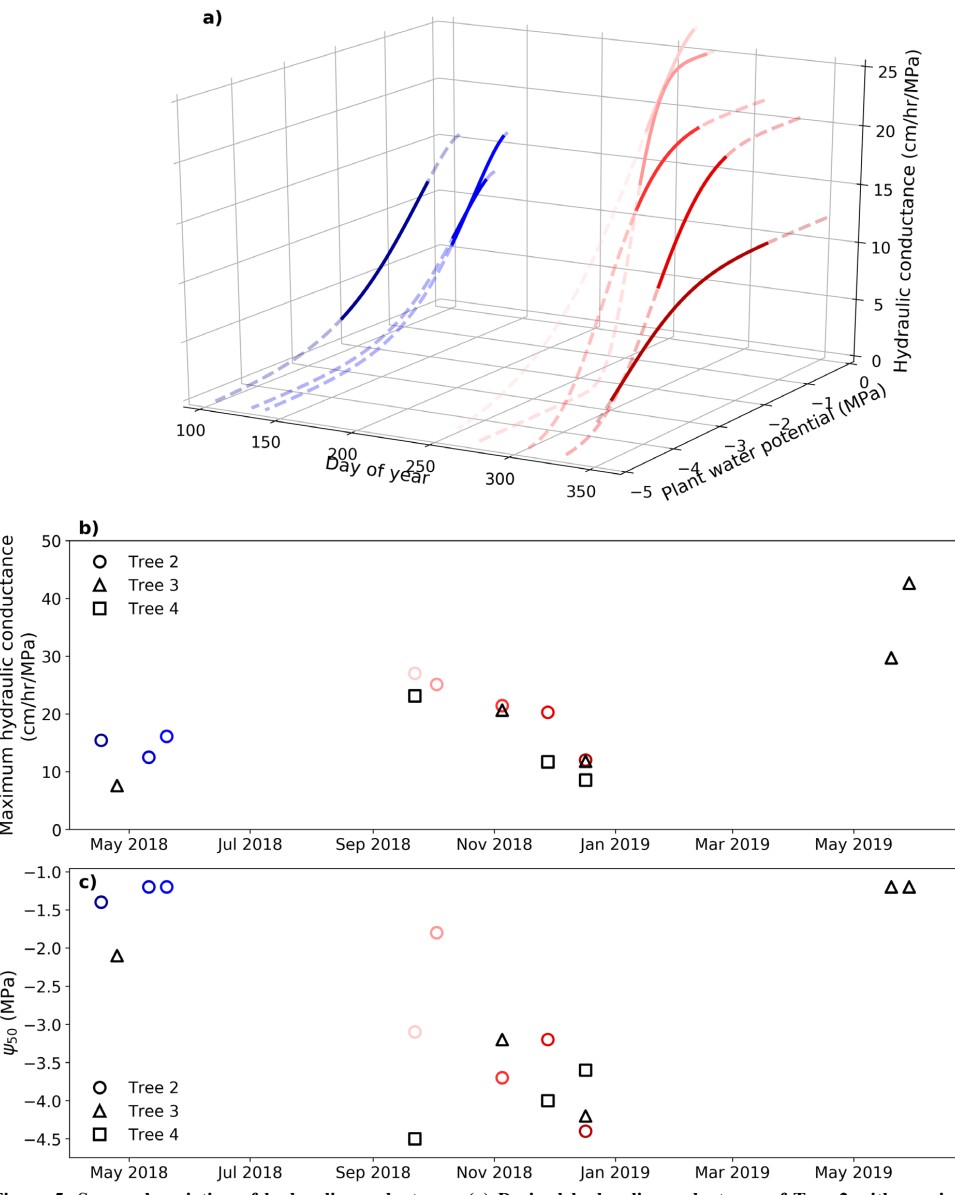

**Figure 5: Seasonal variation of hydraulic conductance. (a) Derived hydraulic conductance of Tree 2 with varying plant water potentials over the calibration periods. The dashed lines show water potentials from -5 to 0 MPa, while the solid lines show the water potentials measured during the calibration periods. (b) The seasonal variation of the derived maximum hydraulic conductance ($k_{max,25}$) of all the trees. (c) The seasonal variation of the derived Ψ50 of all the trees.**

The maximum hydraulic capacitance followed a similar pattern to the maximum hydraulic conductance, with higher values in spring and lower values in autumn. It gradually decreased with an intensification of water stress (Fig. 6a). However, there is an interesting pattern: in November and December, when water stress was



developing, the maximum hydraulic capacitance was relatively small, but the hydraulic capacitance was high where the water potential was low (about -3 MPa; the dark red lines in Fig. 6a). A similar pattern existed for the

other two trees. The seasonal changes in plant hydraulic capacitance and conductance, derived by the pumping-test analogue, indicated that in the wet season (with plenty of rainfall but low transpiration demand), the plants mainly transpired water taken up directly from the soil, and their internal storage was not fully used. In contrast, during the dry season, a greater proportion of the stored water was used to meet the plant transpiration demand. This suggests that the water storage of trees is more important for transpiration at low water potentials during

the dry season than at higher water potentials in the wet season.. This suggests that the effective hydraulic capacitance is not only determined by their intrinsic properties but also by hydrometeorological conditions, that plant effective hydraulic capacitance differs significantly during dry and wet seasons. We conclude these changes of effective hydraulic capacitance from a model-inversion perspective; more direct measurements of plant water storage across seasons are needed to validate these findings.

The pumping-test analogue, based on a whole-plant approach to derive plant hydraulic properties from the relationship between plant water use and water status, differs significantly from traditional laboratory-based measurement methods in the plant physiology community. Despite this distinction, we arrived at the same conclusion as seen in previous studies: plant hydraulic properties exhibit seasonal plasticity, and we can derive more continuous data to support this conclusion. The specific seasonal changes in hydraulic conductance that

we derived ($k_{max,25}$ in Fig. 5b and $\Psi_{50}$ in Fig.5c) align with most existing studies. While the seasonal plasticity of maximum effective hydraulic capacitance ($C_{max}$) is reported for the first time here, the results make sense because the release and recovery of stored plant water are also dependent on the hydraulic conductivity between xylem and storage, which varies seasonally (Hölttä et al., 2006). Additionally, seasonal patterns in xylem and phloem growth, as well as transpiring leaf area, may also influence the release and recovery of stored water,

though these factors remain unknown for this species.

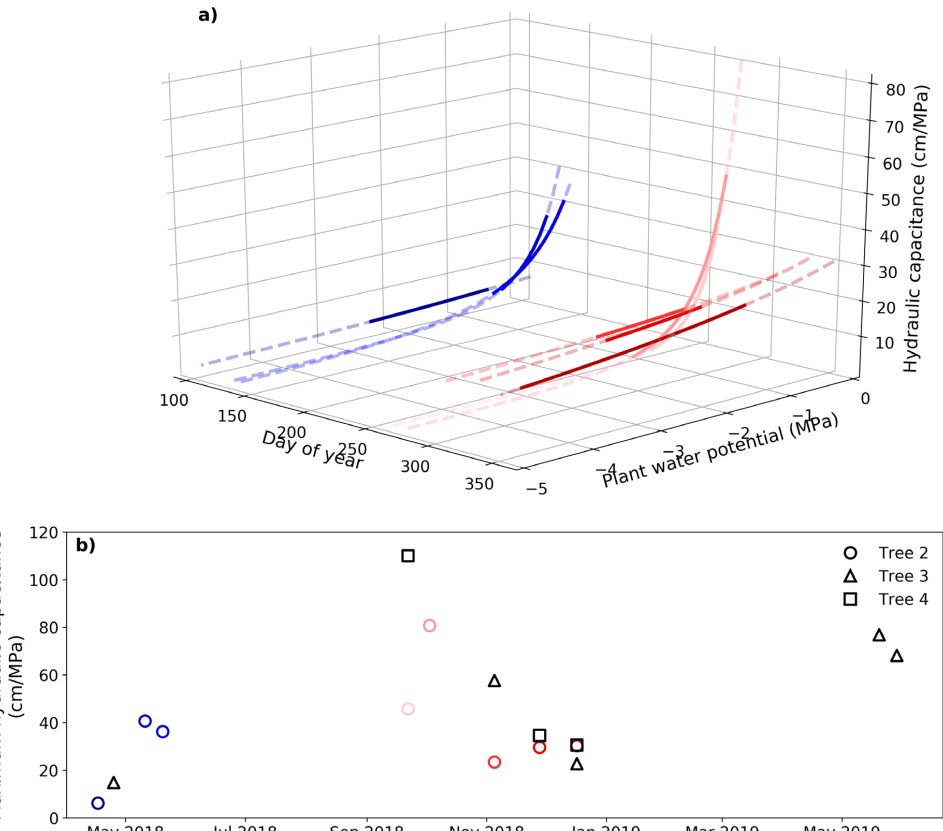

**Figure 6: Seasonal variation of hydraulic capacitance. (a) Derived hydraulic capacitance of Tree 2 with varying plant water potentials over the calibration periods. The dashed lines show water potentials from -5 to 0 MPa, while the solid lines show the water potentials measured during the calibration periods. (b) The seasonal variation of the derived maximum hydraulic capacitance ($C_{max}$) of all the trees.**

### 3.3 Insights from the derived time-varying plant hydraulic properties

By applying calibration and validation methods during dry spells, we identified that the pumping-test analogue is capable of deriving key plant hydraulic properties and reveal that these properties exhibit seasonal plasticity. Building on this result, we explored the plant hydraulic properties near-continuously. Figure 6 presents the plant hydraulic properties ($k_{max,25}$, $\Psi_{50}$, and $C_{max}$) derived using the dynamic-window method. After filtering (calibration NSE > 0.7), the time range of the representative plant hydraulic properties is consistent with the range obtained from the no-rain periods. Except for the period from January to April 2019, most other periods pass the filter criteria. This further supports our previous conclusion that the failure of the method for this period was not incidental. During other periods, the seasonal variation trend of plant hydraulic properties is consistent with the previous results shown in Fig. 5. Maximum hydraulic conductance $k_{max,25}$ decreased during the dry season, especially from October to December 2018 (as shown in Fig. 7), while $\Psi_{50}$ became more negative as the dry season progressed. Figure 7 also shows the variation in root-zone water potential, indicating that changes



in $k_{max,25}$ and $\Psi_{50}$ closely correlate with changes in root-zone water potential. During the dry season, both $k_{max,25}$ and $\Psi_{50}$ decreased as the root-zone gradually dried out; with replenishment from rainfall or irrigation, these values increased accordingly. Moreover, the increase in plant hydraulic properties slightly lagged the recovery of root-zone water potential (Fig. 7), suggesting that plants require time to respond to root-zone moisture replenishment. This phenomenon is reasonable and further validates the reliability of the derived plant hydraulic properties.

As for $C_{max}$, its variation generally follows the trend shown in Figure 6b, where it was high in spring, decreased as summer progresses, stayed low in autumn, and likely gradually increased through winter to spring. As mentioned earlier, effective hydraulic capacitance is influenced not only by the plant water status but also by the atmospheric water demand. It peaked in spring when both root-zone water supply and atmospheric demand for transpiration was high. In winter, despite abundant water supply, low water demand resulted in a relatively small maximum effective hydraulic capacitance, as transpiration primarily relies on water uptake from the roots, with limited contribution from stored water. As water demand increased in spring, $C_{max}$ rose. As the dry season progressed, the root zone moisture was depleted, leading to a reduction in $C_{max}$. The peak observed in January 2019 (inset in Fig. 7c), after an irrigation event, was due to a rapid replenishment of root-zone water combined with high daytime transpiration demand. At night, the plants absorbed significant amounts of water from the root zone into their storage, and with high daytime transpiration demand, the refilled water was quickly transpired during the day. This combination resulted in an exceptionally high $C_{max}$ value.



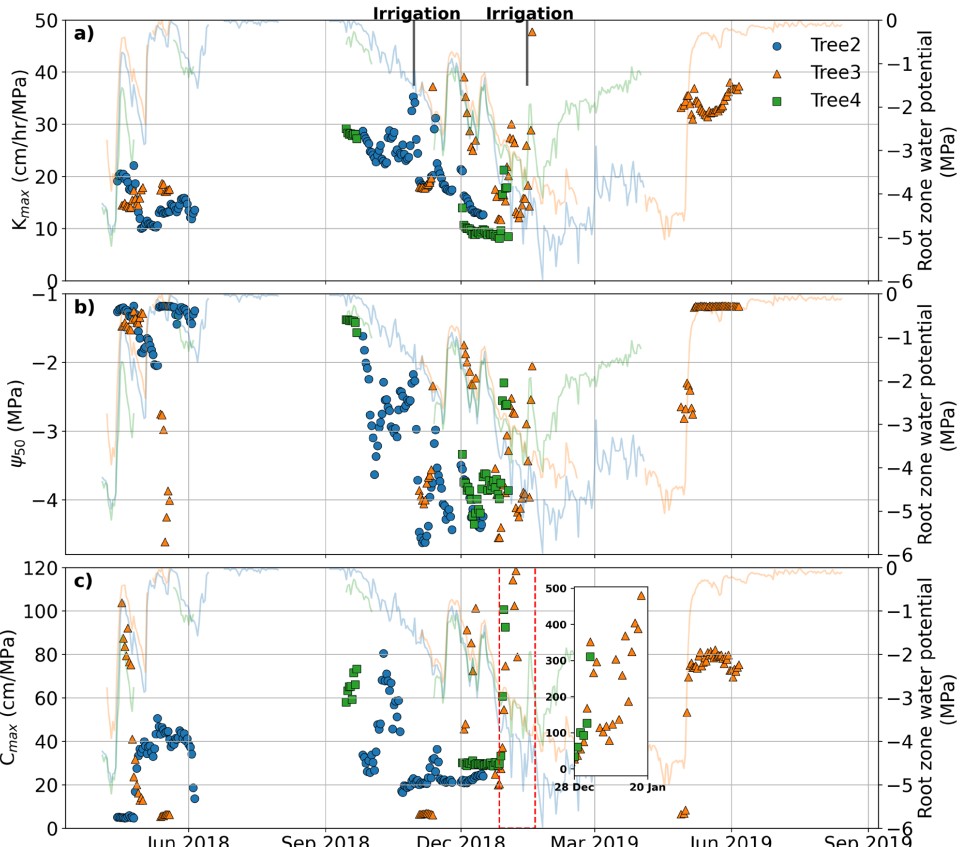

**Figure 7: Derived hydraulic properties from April 2018 to July 2019. (a) Maximum hydraulic conductance, (b) $\Psi_{50}$, and (c) maximum effective hydraulic capacitance were derived based on 20-days calibration periods and only those properties that pass the validation criteria are presented here. The inset in panel (c) highlights the rapid increase in $C_{max}$, reaching 500 cm/MPa following irrigation. The data points represent the midpoint of each 20-day period. The lines show the root-zone water-potential dynamics.**

### 3.4 Relations between seasonal variations of plant hydraulic properties and climate variables

The pumping-test analogue has provided a large amount of plant hydraulic property data, which enables analyses and research that were previously difficult to conduct. This paper presents two preliminary attempts to apply the pumping-test analogue. The first attempt is to explore the relationship between plant hydraulic properties and climate variables. Plant traits and growth are closely linked to climate conditions, with the primary climate factors being moisture, radiation, and temperature, and the key climate variables differing depending on the plant species and the climatic zone (Nemani et al., 2003; Seddon et al., 2016). The correlation between hydraulic properties and climate variables for drooping sheoak under a Mediterranean climate is shown in Table 3. Precipitation, radiation, and temperature can partially explain the seasonal variation in plant hydraulic properties, with the best regression observed for $\Psi_{50}$ explaining 61% of the variation. Precipitation is a significant predictor variable in the regression models for plant hydraulic properties in all three trees, and the coefficient is positive, indicating that plant hydraulic properties increase as moisture conditions improve. This finding aligns with the trend observed in Fig. 7, where plant hydraulic properties and root zone water potential




exhibit similar changes. This suggests that, under a Mediterranean climate, moisture conditions are the primary determinant of the drooping sheoak's hydraulic property variations. Regarding $k_{max,25}$, temperature is a significant predictor with a negative coefficient. Radiation is not a significant predictor in the regression. In Mediterranean climates, temperatures are low in the wet season and high in the dry season, so high temperatures during the dry season exacerbate the reduction in $k_{max,25}$. In terms of $\Psi_{50}$, radiation is a predictor with a negative coefficient, and temperature has a positive coefficient. This suggests that high radiation during the dry season causes a reduction in $\Psi_{50}$, leading plants to adopt a more conservative hydraulic strategy, while temperature has the opposite effect, promoting an increase in $\Psi_{50}$. In contrast, no significant relationships between $C_{max}$ and either temperature or radiation are found by the regression. It is important to emphasize that this is a preliminary trial, based on a limited monitoring duration from a single species within a specific climate zone. The relationship between plant hydraulic properties and climate variables is complex and requires further analysis using longer-term sap-flow and stem water-potential data, as well as additional related data, such as phenology. Additionally, there is a significant correlation between the three climate variables selected in this study, and future research should aim to isolate the independent effects of each climate variable on hydraulic properties.

**Table 3: Multiple linear regression models for the relationship between plant hydraulic properties and climate variables.**

|  | $P_{105}$ | $R_{20}$ | $T_{20}$ | Intercept | $R^2$ |
|---|---|---|---|---|---|
| $k_{max,25}$ | **0.038** | 0.053 | **-1.23** | 37.18 | 0.27 |
| (p-value) | 0.000 | 0.77 | 0.001 | 0.000 | |
| $\Psi_{50}$ | **0.0055** | **-0.21** | **0.14** | -2.51 | 0.61 |
| (p-value) | 0.001 | 0.000 | 0.000 | 0.000 | |
| $C_{max}$ | 0.15 | -0.59 | 1.97 | 6.64 | 0.02 |
| (p-value) | 0.09 | 0.55 | 0.32 | 0.82 | |

\* *Note: Each model's first row shows the regression coefficients (slopes), while the second row displays the corresponding P-values for each climate variable. The intercepts and R² values are also presented.*

**3.5 Seasonal variation of the trade-off between hydraulic efficiency and safety**

The second attempt to apply the pumping-test analogue is to explore the relationship between plant hydraulic efficiency and safety. In theory, higher hydraulic efficiency is associated with relatively less-safe hydraulic strategies. A global metanalysis (Gleason et al., 2016) confirmed this trade-off between hydraulic efficiency and safety across the world's woody plant species, but it is not strong. Trade-offs have also been observed between different parts of a plant within the same tree (Domec et al., 2006; Meinzer et al., 2010). In this study, we attempted to investigate whether a trade-off between hydraulic efficiency (represented by $k_{max,25}$) and safety (represented by $\Psi_{50}$) exists within individual plants across different seasons, as we know that plant hydraulic efficiency and safety can vary seasonally. Figure 8 illustrates a relationship between $\Psi_{50}$ and $k_{max,25}$: as the dry season progressed, plant hydraulic efficiency declined, and the hydraulic strategy shifted to be more conservative and safety-oriented (with $\Psi_{50}$ tending to more negative values). In contrast, with the start of the rainy season, hydraulic efficiency improved, and the hydraulic strategy shifted to a more efficiency-oriented mode (with $\Psi_{50}$ shifting to less negative values). This pattern suggests a potential trade-off between hydraulic efficiency and safety within individual plants. It is noteworthy that although the $\Psi_{50}$ values in May 2018 (circled in Fig. 9) were similar to those in the same period of 2019, $k_{max,25}$ in 2018 was much lower than in 2019. Two possible explanations for this difference are: first, the plant experienced significant sapwood growth from 2018




to 2019, resulting in improved hydraulic efficiency; second, the plant experienced higher water stress during 2017-2018 compared to 2018-2019. This second explanation is supported by Figure 4c, which shows that cumulative rainfall in May 2018 was approximately 50 mm, compared to around 100 mm in May 2019. The lower rainfall in 2018 may have led to more severe cavitation, resulting in a delayed recovery in hydraulic efficiency at the start of the rainy season. This result is further supported by the stem water potential in April-

May 2018, which was lower than during the same period of 2019.

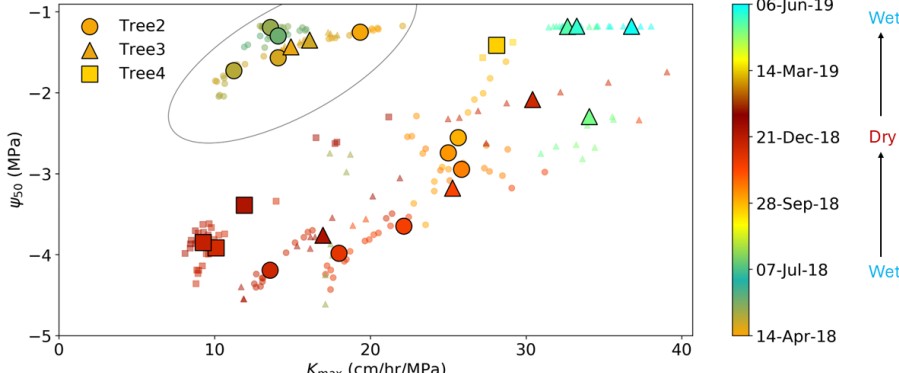

**Figure 8: Relationship between $\Psi_{50}$ and $k_{max,25}$ for individual trees. Small dots represent hydraulic properties derived from 20-day dynamic windows, while larger symbols indicate the mean values of every ten consecutive points to clarify overall trends. The oval includes the points from autumn 2018 and the explanations for the deviations of**

**these points are provided in Section 3.5.**

Based on the above results (Fig. 5-8), Figure 9 illustrates a generalized schematic of plant hydraulic responses in drooping sheoak to water supply and demand under a Mediterranean climate. This schematic summarizes the expected seasonal variations in key hydraulic properties in response to changes in water availability and atmospheric demand. Water supply (blue line) and water demand (orange line) represent the typical seasonal

dynamics in a Mediterranean climate, where water supply is highest during wetter seasons and decreases as the dry season progresses, while water demand follows an inverse pattern, peaking during warmer months. In response to these environmental drivers, the lower three curves illustrate the seasonal dynamics of plant hydraulic efficiency, safety, and buffer. In this study, plant hydraulic efficiency, safety, and buffer are represented by $k_{max,25}$, - $\Psi_{50}$, $C_{max}$, respectively. Plant hydraulic efficiency generally followes changes in water

supply, and plant hydraulic safety shows the opposite pattern—higher hydraulic efficiency corresponds to lower hydraulic safety (Fig. 8). While plant hydraulic buffer (represented by $C_{max}$) responds to both water supply and demand. When either supply or demand is low, plant hydraulic buffer remaines limited. In contrast, it peaks when both supply and demand are high. This genralized schematic help illustrate how hydraulic properties shift in response to seasonal variations in water supply and demand, providing insights into plant water-use strategies

under a Mediterranean climate.



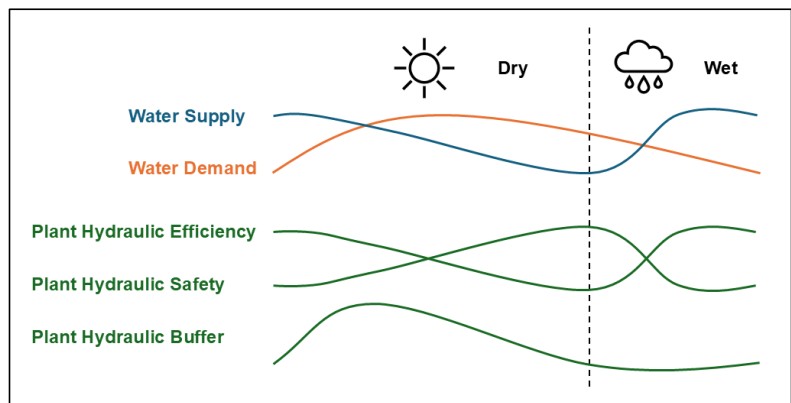

**Figure 9: Generalized schematic of plant hydraulic responses in drooping sheoak (*Allocasuarina verticillata*) to water supply and demand under a Mediterranean climate. In this study, plant hydraulic efficiency, safety, and buffer are represented by $k_{max,25}$, $-\Psi_{50}$, $C_{max}$, respectively. The dry season in Adelaide is from September to April, which is longer than the wet season.**

**3.6 Opportunities arising from the pumping-test analogue**

The two attempts to apply the pumping-test analogue in section 3.4 and 3.5 highlight the future application potential of the pumping-test analogue. Compared with the current laboratory methods, the pumping-test analogue estimates whole-plant parameters, which can be directly applied in models. The method is based on automatic monitoring data, specifically stem water potential and sap flow, and the equipment only needs to be installed once and maintained regularly, which greatly reduces the workload. In addition, the pumping-test analogue is non-destructive, reducing the damage to the plant and the differences caused by sampling on different branches. Additionally, the increasing number of plant water-potential and sap-flow datasets make the application of the pumping-test analogue increasingly feasible.

The near-continuous plant hydraulic property predictions provided by the pumping-test analogue can be used for plant hydraulic research that was previously limited by data availability. It also offers a new possibility for parameterizing plant hydraulic processes in land-surface models. As mentioned in the introduction, seasonal variations of plant hydraulic properties have been reported within the plant physiology community; however, its application in hydrological and land-surface modelling is still rare. The pumping-test analogue could serve as a potential approach to represent and parameterize the seasonal plasticity of plant hydraulic properties in numerical hydrological models.

In short, the pumping-test analogue method is easy to implement, making it suitable for estimating plant hydraulic parameters and their ranges, with potential application to hydrologic models. Due to its easy implementation, it is possible to use the pumping-test analogue at different locations or on different species to conduct spatial and inter- and intra-species analysis, which may allow us to build a comprehensive understanding of plant hydraulic property variation.

**4 Conclusions**

We developed a new, in-situ method to derive representative whole-plant hydraulic properties through a pumping-test analogue. We developed and applied the pumping-test analogue method on *Allocasuarina*



*verticillata* and revealed the seasonal variation of maximum hydraulic conductance, effective capacitance, and hydraulic vulnerability. The hydraulic properties obtained through this method from one-week monitoring is suitable for modelling plant hydraulic processes for at least one month, excepet during the periods of severe water stress. This method provides new evidence for the seasonal plasticity of plant hydraulic properties and provides insights into how the hydraulic properties in drooping sheoak vary seasonally under Mesiterranean climate.

In this study, we also used near-continuous hydraulic properties derived by the pumping-test analogue to explore the relationship between plant hydraulic properties and climate variables, as well as the trade-off between hydraulic efficiency and safety within individual plants across different seasons. These preliminary findings indicate that is feasible to estimate plant hydraulic properties under various conditions. Consequently, this method enables a comprehensive understanding of the variations in plant hydraulic properties and provides the potential for accurate estimation of plant hydraulic parameters for land-surface process models. An accurate description of plant hydraulics in land-surface models will help decrease the uncertainty of the water and carbon flux simulation.



**Acknowledgments**

Craig Simmons from The University of Newcastle, Australia is greatly appreciated for providing comments and suggestions at an early stage of this research. Zijuan Deng, Yifei Zhou, Na Liu and Zidong Luo assisted with field data collection. Zhechen Zhang appreciates the financial support from the China Scholarship Council.

**Data and code availability**

Software, including Sap Flow Tool (https://ictinternational.com/manuals-and-brochures/sfm1x-sap-flow-meter/),
Matlab R2012b (https://au.mathworks.com/products/matlab.html), and Python is used to generate data sets, models, and figures. All the data was submitted to PSInet for public access.

**Author contribution**

ZZ - Formal analysis, Methodology, Investigation, Visualization, Writing – original draft preparation; HG – Conceptualization, Investigation, Resources, Supervision, Writing – review & editing; EV – Methodology,
Writing – review & editing; KS – Methodology, Writing – review & editing; OB - Writing – review & editing

**Competing interests**

The authors declare that they have no conflict of interest.

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
