# Peer review of "Revealing Seasonal Plasticity of Whole-Plant Hydraulic Properties Using Sap-Flow and Stem Water-Potential Monitoring"

_EGUsphere, 2025_

## Referee Comment (RC2)

Review for Revealing Seasonal Plasticity of Whole-Plant Hydraulic Properties Using Sap-Flow and Stem Water-Potential Monitoring

**Summary**

Zhang and colleagues present an innovative study that generated novel data and interpreted them within an effective modeling framework, demonstrating substantial seasonal variation in plant hydraulic properties that are often assumed to be static. I appreciate the clear writing and consider this manuscript an important addition to literature. However, I believe that additional care in comparing standard vs. new approaches and embedding the ecophysiological literature will help this work reach a wider audience.

**Major comments**

Given that this is a new approach, one inevitable question is how Kmax and P50 derived from this method compares to existing, branch-harvest approaches. Are there any published literature values that speak to the range of values found in this study?

Another inevitable question is how confident can we be in values derived from short periods of data, which cannot encompass the entire range of dry downs found in branch-harvest approaches? Using 3D plots in Fig. 5a beautifully demonstrates seasonal variation within Tree 2 but does not reassure readers on this concern. I recommend presenting representative curves, say from the wet and dry seasons, in 2D to show both the raw data and the fitted values, so that readers can judge for themselves whether this approach yields parameters that mimic our conventional understanding of Kmax and P50.

I appreciate the careful field and modeling work to demonstrate time-variant hydraulic properties, especially their correlation with seasonality and hydrometeorological properties. However, the plant's biological responses can be more greatly emphasized. Consider expanding Ln 348-350 by discussing possible direction of relationships and adding citations.

I am excited to see that Kmax and P50 correlate with root zone water potential, suggesting that traits are far more dynamic than typically accounted for in models. But, given that predawns are used to represent root zone water potential and utilized to derive Kmax and P50, is this correlation independent? Also consider characterizing the duration of the lag and speculating on the biological processes that could account for it.

Why were these three climate variables selected? It is well-established that soil moisture/soil water potential and vapor pressure deficit are endmembers of the SPAC and directly impact plant hydraulic transport, and by extension plant hydraulic properties. Consider re-running the regression analyses with more proximal predictor variables that are indicative of moisture conditions. Specifically, in Ln 405-406, these climate variables are referred to collectively as moisture conditions, which does not seem accurate.

Plant phenology often refers to the timing of leaves and flowers, but in this case the timing of vessel development might strongly impact hydraulic properties. Consider further discussion of the kinds of phenological data that will influence plant hydraulics, including vessel development and distribution, leaf area to sapwood area to root area ratios, etc.

It appears P50 was derived from the curve fitting parameters of the Weibull model. Was P50 calculated within the model for each iteration, and then summarized as the posterior mean? If so, what about the measure of spread, such as a central 95% credible interval? I'd like to see these presented as error bars in the figures, which will allow readers to interpret the magnitude of the seasonal variation. Such error bars in both x- and y- directions would be especially helpful in Fig. 8, which can have different meanings for the small (credible intervals) and large symbols (sample standard deviations).

**Minor comments**

Ln 50. Change to "which is likely due to"

Ln 55. Change to "informing the"

Ln 335. Remove the duplicated period.

Ln 366. Subscript the 50.

Ln 451. Change to positive/negative effect, rather than coefficient

Fig. 8 caption. The color axis is not explained – is it simply time of year?

**Suggested reference**

Ogle K, Barber JJ, Willson C, Thompson B. Hierarchical statistical modeling of xylem vulnerability to cavitation. New Phytologist. 2009. 182(2):541-554.

---

## Author Comment (AC1)

We sincerely thank the reviewer for the encouraging and constructive comments. We appreciate your positive assessment of our study, particularly the potential of our approach to contribute to the development of standard methods for deriving plant hydraulic properties in a non-destructive manner.

Below, we provide point-by-point responses to the reviewer's specific comments.

**1. *Eq6: where is this equation used?**

Equation 6 is used to adjust the influence of temperature on hydraulic conductance, allowing the derived values to be standardized to a reference temperature of 25 °C. Specifically, Eq. 6 is substituted into Eq. 5 to compute the whole-plant hydraulic conductance at the corresponding temperature and water potential. All subsequent hydraulic conductance values presented in the manuscript—including those in Table 1—are reported at 25 °C. We will improve the description of this part in the revised manuscript.

**2. *L169: add country name of Flint University.**

Flinders University is in Adelaide, Australia. We will add it in the revised manuscript.

**3. *Fig3: unit of daily radiation is MJ/m2/day; unit of daily precipitation is mm/day**

We will change it in the revised manuscript.

**4. *Fig 3c: how can the accumulative precipitation go down? Do you mean net precipitation here? e.g., precipitation minus (actual/reference/potential) evaporation? In case of the latter, clearly explain which evaporation is used.**

Thank you for pointing out this confusion. In Fig. 3c, the "accumulative precipitation" actually refers to the sum of rainfall over the preceding 105 days, calculated on a rolling basis. Since this is a moving window sum rather than a cumulative total from the start of the season or year, it can decrease if recent rainfall decreases compared to earlier periods within the window. We agree that the current label may be misleading, and we will directly describe it as the sum of rainfall in the preceding 105 days. We are not referring to net precipitation or subtracting any form of evaporation in this figure.

**5. *Fig4: this figure is rather small and therefor difficult to read.**

Thank you for the helpful comment. We agree that the current layout makes Fig. 4 difficult to read. In the revised manuscript, we will change the layout to a one-column format to increase the figure size and improve legibility. The revised version of the figure is shown below.

[Figure]

6. *Fig4: In your validation periods you largely underestimate sapflow in summer. This is also the case when you calibrate on summer sapflow. How is this possible? You'll expect that the parameters would be able to simulate this? I think it will be good to elaborate on this mismatch in the manuscript. Furthermore, could equifinality in model parameters play a role here? No information on parameter optimization is shared.*

Thank you for this insightful comment. We agree that underestimation of sap flow occurs in some summer calibration periods (notably P5, P6, & P7) , and we will elaborate on this mismatch in the revised manuscript. Specifically, we will add the following explanation:

*"…… In some early summer calibration periods, the model underestimates sap flow in late summer (P5 - P7). This underestimation is not due to a limitation in maximum hydraulic conductance but rather results from the shape of the vulnerability curve, specifically the relatively high (less negative) $\Psi_{50}$ values derived for these periods. A higher $\Psi_{50}$ indicates greater sensitivity of hydraulic conductance to declining water potential, causing hydraulic conductance to drop rapidly under moderate water stress. This sharp decline in hydraulic conductance effectively limits sap flow in the simulation. The temporal dynamics of $\Psi_{50}$ are analysed in more detail in the next section (see Fig. 5)……"*

Regarding the information on parameter optimization, the optimal parameter values are presented in Figure 5 to 7, and their posterior distribution are shown in Figure S3. These distributions are tightly clustered and exhibit narrow ranges, suggesting relatively low uncertainty. We will revise Figure 8 to additionally display the standard deviations of the posterior parameters, in line with Reviewer #2's suggestion. While equifinality is always a consideration in model calibration, in this case, we believe its impact is limited. The narrow posterior distributions indicate that the model parameters are relatively well-constrained and that equifinality is unlikely to be a major driver of the observed mismatch.

7. *Fig 4: Currently, you calibrated on a few days. What if you would calibrate on an entire season? Would the model then be better in capturing all-season plant hydraulic properties?*

Thank you for this thoughtful question. Based on your suggestion, we tested two additional calibration strategies to explore this idea further. First, we calibrated the model over the entire data period (see Fig. R1). The results show that sap flow is overestimated in the winter and underestimated in the summer. This suggests that plant hydraulic properties vary seasonally, and calibrating over the whole year leads to parameter values that represent a compromise between winter and summer conditions. As a result, the model fails to capture seasonal dynamics accurately. Second, we calibrated the model using only summer data (September to December), and the results are shown in Fig. R2. As expected, the model performs well during the calibration period (summer), but significantly overestimates sap flow during the winter. This again highlights the seasonal variability in hydraulic properties. In summary, calibrating over an entire season can effectively capture the average behaviour during that season, but it cannot fully represent plant hydraulic properties across all seasons due to their temporal variability. To accurately capture seasonal dynamics, it is necessary to allow model parameters

to vary over time. We will include Figs. R1 and R2 in the Supplement Materials to support this discussion.

[Figure]

Figure R1. Simulated sap flow from the whole data period compared with observed sap flow (NSE = 0.54).

[Figure]

Figure R2. Simulated sap flow from the whole data period compared with observed sap flow (Calibrated NSE = 0.65; validated NSE = 0.37).

**8. *Table2: From this table it seems that Tree2 is 'the best performing' tree. How do the sapflow time series of the other trees look like?***

The sap flow time series of the other trees in calibration and validation are provided in Figs. S1 and S2 (in the same format as Fig. 4) in the Supplementary Materials. Their performance is similar to that of Tree 2. We chose to present Tree 2 in the main text primarily due to space limitations and the fact that it has the largest number of calibration periods.

**9. *Fig5+6: What is the difference between blue and red?***

Thank you for pointing this out. In Figures 5 and 6, we use colour to indicate seasonal context: cool colours (e.g., blue) represent the wet season, while warm colours (e.g., red) represent the dry season. We will clarify this in the revised figure captions to avoid confusion.

**10. L359: Figure 6 => Figure 7?**

Yes, thank you for catching this. We will correct the figure reference from Figure 6 to Figure 7 in the revised manuscript.

---

## Author Comment (AC2)

**Summary**

Zhang and colleagues present an innovative study that generated novel data and interpreted them within an effective modelling framework, demonstrating substantial seasonal variation in plant hydraulic properties that are often assumed to be static. I appreciate the clear writing and consider this manuscript an important addition to literature. However, I believe that additional care in comparing standard vs. new approaches and embedding the ecophysiological literature will help this work reach a wider audience.

**Reply:** We sincerely thank the reviewer for their insightful and encouraging comments. We are especially grateful for the suggestions to strengthen the comparison between our approach and standard laboratory methods, and to further embed our findings within the broader ecophysiological literature. We believe these points are crucial for enhancing both the scientific rigour and the accessibility of our manuscript. In response, we will carefully revise the manuscript to clarify the advantages and limitations of our approach and expand the discussion of biological mechanisms underlying the observed seasonal plasticity of hydraulic properties. We are confident that these improvements will make the manuscript more robust and relevant to a wider audience. Detailed responses to each comment are provided below.

**Major comments**

**Given that this is a new approach, one inevitable question is how $K_{max}$ and $P_{50}$ derived from this method compares to existing, branch-harvest approaches. Are there any published literature values that speak to the range of values found in this study?**

**Another inevitable question is how confident can we be in values derived from short periods of data, which cannot encompass the entire range of dry downs found in branch-harvest approaches? Using 3D plots in Fig. 5a beautifully demonstrates seasonal variation within Tree 2 but does not reassure readers on this concern. I recommend presenting representative curves, say from the wet and dry seasons, in 2D to show both the raw data and the fitted values, so that readers can judge for themselves whether this approach yields parameters that mimic our conventional understanding of Kmax and P50.**

**Reply:**

We appreciate the reviewer's thoughtful comments and agree that it is important to compare our approach with existing branch-harvest approaches. However, it is important to emphasize that the parameters estimated by our method differ in context from those obtained via traditional branch-harvest methods. Specifically, our approach yields whole-plant hydraulic properties rather than properties of a small section of the plant, and the values from our approach reflect the effective hydraulic behaviour of plants under field conditions. In contrast, traits measured in the lab often reflect the inherent structural properties of plant tissues under controlled conditions, part of which may never happen in the field condition for the season of the branch harvesting.

More specifically, there are several challenges with comparing branch-level to whole plant. For example, K declines with stem diameter due to the increasing amount of non-functional xylem in older stems. $P_{50}$ is likely to differ in distal branches compared to main stems. More importantly, whole-plant K and $P_{50}$ include the root system, which is well known to have quite different hydraulic properties to the aboveground parts, and the literature simply doesn't have data for these. For K, it would make more sense to compare with published whole-plant conductance. The $P_{50}$ is less scale-dependent, but very much tissue-dependent (roots and stems of different diameter/age/position). Thus, while comparison with branch-harvest methods should not be seen as a validation of our approach—due to fundamental differences in scale and physiological scope—such comparisons can still offer valuable context. They help assess whether our derived parameters fall within reasonable ranges.

In the original manuscript, we did not include such a comparison mainly due to the limited availability of hydraulic trait data for our species — drooping sheoak (*Allocasuarina verticillata*). Neither the published literature nor public databases (e.g., TRY) contain records of vulnerability curve parameters such as $K_{max}$ or $P_{50}$ for this species. To provide a preliminary comparison, we expanded our search to include other species in the *Allocasuarina* genus, as well as closely related species in the *Casuarina* genus. Through this effort, we found only one relevant record in the TRY database: a vulnerability curve for *Allocasuarina campestris*, with a reported $P_{50}$ value of $-2.96$ MPa. This value falls within the range of $P_{50}$ values we estimated using our approach ($-1.2$ MPa to $-4.5$ MPa). We present this comparison in Fig. R1, which shows the measured curve alongside the seasonal variation of our inferred vulnerability curves. Notably, the published value lies near the centre of our seasonal range and is also very close to the curve we obtained using the whole data period as the calibration period, which lends some support to the validity of our parameter estimates. Given the taxonomic and methodological differences involved, we won't present Fig. R1 in the main text but will include the corresponding discussion on the measured value mentioned above in Section 3.2 of the main text. This will help readers better understand the applicability and limitations of our method and strengthen confidence in our results.

As for changing Figure 5a to 2D to show the raw data and fitted values, we don't have raw data on the vulnerability curve, as the curve is fitted from the whole-plant hydraulic model. The parameters changes are shown in Figure 5b and Figure 5c.

[Figure]

Figure R1. The lab-based measured vulnerability curve and derived vulnerability curves from pumping-test analogue. The red curve is derived from 17/12/2018 representing the dry period and the blue curve is derived from 17/04/2018 representing the wet period. The dotted curve is derived from all the data to present the mean value for the whole period, and the solid line is the curve derived from the points in TRY dataset (shown by the stars).

**I appreciate the careful field and modeling work to demonstrate time-variant hydraulic properties, especially their correlation with seasonality and hydrometeorological properties. However, the plant's biological responses can be more greatly emphasized. Consider expanding Ln 348-350 by discussing possible direction of relationships and adding citations.**

**Reply:** We fully agree with the reviewer that the plant's biological responses should be more discussed when interpreting the seasonal variation in hydraulic properties, particularly in relation to the maximum hydraulic capacitance ($C_{max}$).

We will modify Ln 345-350 as follows:

*" …… While the seasonal plasticity of maximum effective hydraulic capacitance ($C_{max}$) is reported for the first time here, the results make sense because the release and recovery of stored plant water are also dependent on the hydraulic conductivity between xylem and storage, which varies seasonally (Hölttä et al., 2006). A higher hydraulic conductivity may facilitate more efficient mobilization of stored water, leading to higher effective capacitance. In addition, seasonal structural changes in plants may further influence hydraulic capacitance. During the wet season, the formation of sapwood and bark tissues may expand the plant's internal water storage capacity, thereby affecting capacitance. It should be noted, however, that we currently lack data on seasonal phenological changes—such as xylem growth, phloem development, or variation in leaf area—for drooping sheoak. Therefore, the mechanisms discussed above remain hypothetical and need further investigation…… "*

In a subsequent reply, we also expand on the biological basis for the seasonal plasticity of $K_{max}$ and $P_{50}$, to provide a more comprehensive explanation of how plant hydraulic traits interact with seasonal climate variability.

**I am excited to see that Kmax and P50 correlate with root zone water potential, suggesting that traits are far more dynamic than typically accounted for in models. But, given that predawns are used to represent root zone water potential and utilized to derive Kmax and P50, is this correlation independent? Also consider characterizing the duration of the lag and speculating on the biological processes that could account for it.**

**Reply:** We thank the reviewer for the insightful comments and interest in the relationship between $K_{max}$, $P_{50}$, and root zone water potential.

While predawn root zone water potential was used in deriving $P_{50}$ and $K_{max}$, the key input variables were sapflow and instantaneous stem water potential (hourly data as inputs). The impact of predawn water potential in the derived $P_{50}$ and $K_{max}$ is considered to be small. Thus, the resulted relationship between $K_{max}$ ($P_{50}$) and root zone water potential is deemed reliable. This is also supported by the fact that the seasonal variation of the relationship can be interpreted reasonably. To avoid the potential bias, we deliberately excluded root zone water potential as a variable in the multiple linear regression analyses that aimed to identify climatic drivers of hydraulic plasticity. This was to avoid introducing statistical dependence between explanatory and response variables that could bias the interpretation.

Regarding the lag duration, we agree that understanding the time lag between environmental drivers and hydraulic responses could provide important insight into underlying biological processes. However, we cannot robustly estimate lag times in this study. Each data point in Figure 7 represents not a single day, but rather the centre of a 20-day moving window used to estimate hydraulic properties. Thus, the estimated $K_{max}$ and $P_{50}$ values reflect the integrated response over each 20-day period, rather than daily variations. This calculation approach limits our ability to detect short-term lag effects. In future studies with higher temporal resolution (e.g., daily measurements of plant hydraulic properties), we can investigate lag structures more explicitly and link them to specific biological mechanisms, such as xylem growth, root activity, or water storage dynamics.

**Why were these three climate variables selected? It is well-established that soil moisture/soil water potential and vapor pressure deficit are endmembers of the SPAC and directly impact plant hydraulic transport, and by extension plant hydraulic properties. Consider re-running the regression analyses with more proximal predictor variables that are indicative of moisture conditions. Specifically, in Ln 405-406, these climate variables are referred to collectively as moisture conditions, which does not seem accurate.**

**Reply:** We fully agree with the reviewer's point that soil moisture and vapor pressure deficit are the two end-point conditions of the soil–plant–atmosphere continuum and directly influence plant water transport. Therefore, they are physiologically more proximal indicators of plant hydraulic responses.

Here we aimed to interpret seasonable variation of plant hydraulic properties, which is less dynamic than vapour pressure deficit. The seasonal variation of VPD in our climate zone is captured by other climate variables (high correlation between VPD and temperature shown in Figure R2). Root zone moisture data (predawn plant water potential) was used in deriving plant hydraulic properties, as explained above, and therefore excluded as a predictor variable. Thus, we selected precipitation, temperature, and radiation as the climatic variables to explain the seasonal plasticity of plant hydraulic properties because they are among the most considered factors in ecological climatology studies and are known to strongly influence plant physiological processes (Nemani et al., 2003; Seddon et al., 2016). These three variables broadly represent the environmental energy input and water availability and thus hold important ecological significance.

Additionally, we acknowledge that using "moisture conditions" in Lines 405–406 of the original manuscript is not sufficiently accurate. As this phrasing may be misleading, we will revise it to a more straightforward term, such as "precipitation conditions," in the revised manuscript.

[Figure]

Figure R2. Correlation between VPD and temperature. Data is from nearby St Mary Park weather station from 2020 to 2023.

**Plant phenology often refers to the timing of leaves and flowers, but in this case the timing of vessel development might strongly impact hydraulic properties. Consider further discussion of the kinds of phenological data that will influence plant hydraulics, including vessel development and distribution, leaf area to sapwood area to root area ratios, etc.**

**Reply:** We fully agree with the reviewer that the vessel development may have a significant impact on plant hydraulic properties. Incorporating some speculative discussion based on phenology and plant physiology may provide a valuable perspective in our study. However, a

proper analysis of these mechanisms would require dynamic measurements of growth and vessel traits across different seasons—data that are currently unavailable in public datasets. For the species of this study, Allocasuarina verticillata, there is a lack of detailed phenological information in the existing literature and datasets. As a result, we are unable to directly analyse seasonal patterns in vessel development.

Nevertheless, some physiological studies have proposed insightful hypotheses. For example, as summer drought intensifies, plants may produce more cavitation-resistant vessels, while in the wetter winter months, they may form larger, more conductive but more vulnerable vessels—thus maximizing hydraulic efficiency and growth. Therefore, it is reasonable to speculate that vessel development and repair processes are closely linked to seasonal variation in hydraulic properties. Also, in a population of xylem vessels, the most vulnerable ones will cavitate first, leaving behind a reduced number (lower K) with greater resistance to cavitation (more negative $P_{50}$). Another hypothesis is that seasonal shifts in plant hydraulic properties may be influenced by changes in xylem sap composition. Recent research has shown that lipids present in xylem sap can alter the sap's surface tension (Schenk et al., 2018; Yang et al., 2020), which may potentially affect hydraulic efficiency and safety. While these hypotheses remain to be directly tested, they offer potential directions for understanding the mechanism of the seasonal plasticity of plant hydraulics. Future research combining detailed physiological measurements with modelling efforts could help elucidate the biological mechanisms underlying these patterns.

We will incorporate this discussion into the revised manuscript to provide a more comprehensive interpretation of the seasonal variability in plant hydraulic properties.

**It appears P50 was derived from the curve fitting parameters of the Weibull model. Was P50 calculated within the model for each iteration, and then summarized as the posterior mean? If so, what about the measure of spread, such as a central 95% credible interval? I'd like to see these presented as error bars in the figures, which will allow readers to interpret the magnitude of the seasonal variation. Such error bars in both x- and y-directions would be especially helpful in Fig. 8, which can have different meanings for the small (credible intervals) and large symbols (sample standard deviations).**

**Reply:** Yes, in this study, the $P_{50}$ values were derived from the fitted parameters of the Weibull model but only based on the optimal parameter set for each calibration period. We fully agree with the reviewer's suggestion that including error bars would help readers better interpret the magnitude and significance of seasonal variations in hydraulic properties.

In the revised manuscript, we will include error bars in both the x- and y-directions in Fig.8 to reflect the uncertainty in parameter estimation. However, we would like to note a potential concern: Figure 8 contains approximately 200 data points, and adding error bars for each point may make the figure cluttered and difficult to interpret. To address this, we propose a compromise. If the error bars can be clearly displayed without affecting readability, we will include them directly in the main figure. Otherwise, we will present the full version with error bars in the supplementary material and retain a simplified version in the main text to balance clarity and completeness.

**Minor comments**

**Ln 50. Change to "which is likely due to"**

**Ln 55. Change to "informing the"**

**Ln 335. Remove the duplicated period.**

**Ln 366. Subscript the 50.**

**Ln 451. Change to positive/negative effect, rather than coefficient**

**Fig. 8 caption. The color axis is not explained – is it simply time of year?**

**Reply:** We thank the reviewer for the careful reading and helpful suggestions. We will revise the manuscript based on these comments. Regarding the legend in Figure 8, the colour bar represents the continuous date across the full observation period. Warm colours generally correspond to the dry season, while cool colours represent the wet season. It is not a simple mapping to the day of year, as the colour bar was designed to distinguish between 2018 and 2019, hence the slight differences in colour for the same calendar days across years. We will revise the figure caption in the updated manuscript to clarify this for readers.

**Suggested reference**

**Ogle K, Barber JJ, Willson C, Thompson B. Hierarchical statistical modeling of xylem vulnerability to cavitation. New Phytologist. 2009. 182(2):541-554.**

**Reply:** We will cite this paper in an appropriate place in the revised manuscript.